# More than a Man, Less than a Painter: David Smith in the Popular Press, 1938–1966

**Paula Wisotzki**

Department of Fine and Performing Arts, Loyola University Chicago, Chicago, IL 60660, USA; pwisots@luc.edu

**Abstract:** Media coverage was vital in establishing the popular reputation of the Abstract Expressionists. Reporting regularly relied on photographic portraits to present these artists as modernist innovators who were an extension of (or even a replacement for) the work of art. Jackson Pollock came to epitomize the Abstract Expressionist artist, with "action" photographs capturing his radical painting method. Pollock's contemporary, American sculptor David Smith, similarly transformed his medium—in his case by embracing industrial methods to make three-dimensional objects. However, given the constraints inherent in the process of welding he employed, how could Smith's image be reconstituted as a celebration of artistic individuality so crucial to modernism? The very method Smith embraced to push the boundaries of art kept him from representing the genius creator who channeled the forces of nature to produce culture. By tracing photographs documenting his career published in local and regional newspapers, popular magazines from *Popular Science* to *Life*, and mass art magazines from *Magazine of Art* to *Arts*, this paper demonstrates that images of Smith at work as an anonymous industrial worker enveloped in protective gear were regularly balanced with images of contemplation—the traditional image of the artist as mediating intelligence. Yet, over the years of his career, the problem of representing Smith was addressed somewhat differently. Early on, there was a tendency to show Smith applying his novel art-making techniques to the production of more traditional objects. During World War II, when Smith was employed as a commercial welder, Smith the artist legitimized reporting on Smith the worker. Finally, in the post-war world—as Smith benefited from the burst of publicity surrounding the triumph of Abstract Expressionism—his rigorous manipulation of metal was celebrated as masculine display, effectively shifting attention away from common industrial labor to heroic individual struggle.

**Keywords:** Abstract Expressionism; *Life* (Periodical); New York School; sculpture; Smith, David 1906–1965



## 1. More than a Man, Less than a Painter: David Smith in the Popular Press, 1938–1966

Who was David Smith? How did photographs of the artist published during his lifetime define the artist for the public? To what extent was the message of those photographs unique to him, and how much did the modernist context in which they appeared shape the ways in which they were presented and received? Scholarship by Rosalind Krauss, Joan Pachner, and Sarah Hamill has brought needed attention to David Smith's photographs as a meaningful part of his artistic output ([Krauss and Pachner 1998](); [Hamill 2015]()). Prior to these contributions, Smith's photographs—those he took himself and those taken by other photographers—were employed almost exclusively as documentation of Smith's career ([Carmean 1982]()). In other words, they served to establish the date of a work, or to verify different methods he employed to produce sculpture. What has not been sufficiently studied is the role such photographs played in defining Smith as an artist. For this project, I will turn to photographs of Smith published during his lifetime, many of them showing him at work making sculpture. I will argue that their content and presentation served to establish him as a leading American modernist artist.[1] However, at a time when public attention shifted to the process of art making, Smith's dominant medium—welding metal—challenged those

seeking to present him in the mold of other innovative modernist artists who were by and large painters. The reality of his chosen medium meant that documenting Smith's studio practice often resulted in images of an anonymous laborer. While some photographs were employed to explain and defend his process, others served to maintain the notion of the artist as a heroic individual—a trope that has persisted in the West for centuries. By the 1950s, the image of the artist acquired such importance that the photographs of artists at work arguably eclipsed reproductions of finished works of art for they could testify to the process of making so vital to modernism. Despite the challenges presented by Smith's chosen materials and techniques, he was caught up in this powerful trend.

The interest in process and the challenge of celebrating the individual were on full display in 1952 when Smith first appeared on the pages of *Life* magazine (A Sculptor Forges Iron 1952). The article opened with a full-page black and white photograph of him welding (Figure 1). The arresting image offered dramatic contrasts between the fire forming the weld and the white-hot droplets of molten metal splashing to the floor and the shadowy forms of the welder with his equipment. His protective gear eliminated any sense of a specific individual. Only the words "An Artistic Smith at Work" running along the bottom of the page suggested the identity of the figure and established a connection to art.[2]

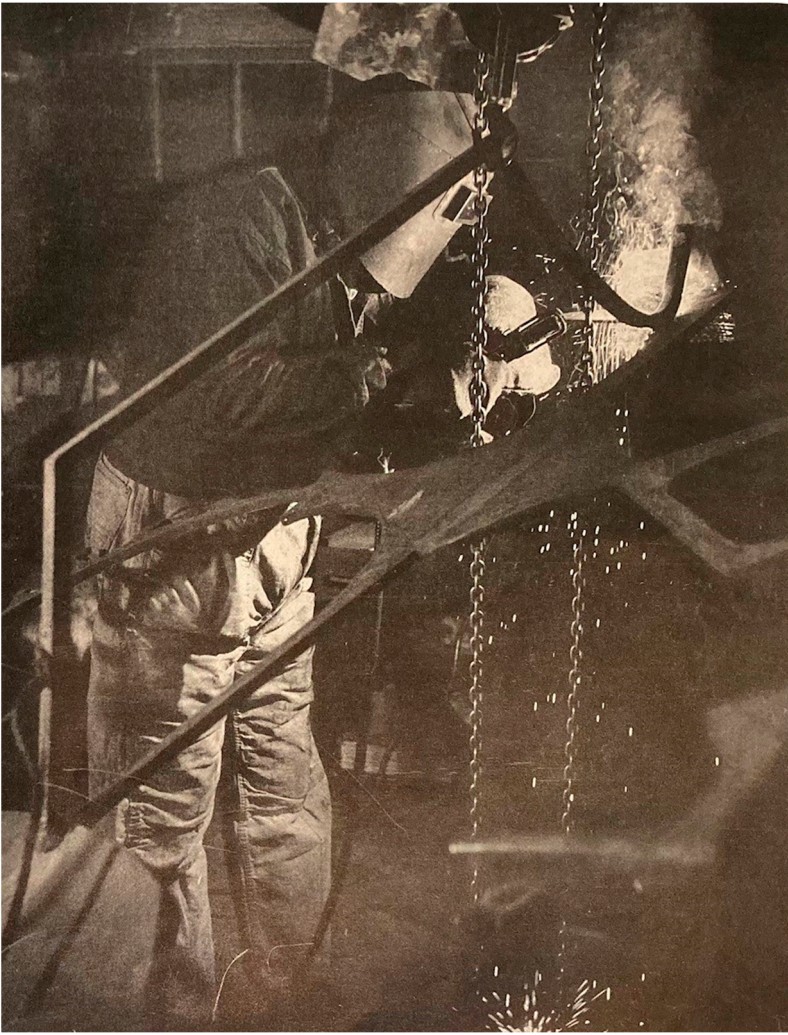

**Figure 1.** David Smith welding in his Bolton Landing workshop, 1952. Photograph by John Stewart.

When the reader turned the page, he was presented with a more conventional portrait of the artist (Figure 2). There, still wearing the welding helmet but with its faceplate raised, Smith was shown from the shoulders up. Stating the obvious, the caption read

"With mask lifted, Smith looks like this". This second image was all the more intimate and revealing because it was presented in contrast to that of the previous page. Its greater accessibility was further acknowledged by the headline "His first name is David". By pairing these two photographs, *Life* assimilated the image of Smith making art, a depiction redolent with the idea of an industrial worker, into a celebration of artistic individuality. The combination of these two photographs acknowledged and attempted to defuse any tension between the image of the anonymous figure employing commercial techniques to weld metal and the notion of the creative individual as worth knowing. The few sentences of text accompanying *Life*'s photographs of Smith further served to legitimize the artist's borrowing of industrial methods as appropriate for making art. They also fed the trope of the underappreciated artist, describing his sculpture as having "earned him tremendous prestige" but having "brought only a meager income".[3]

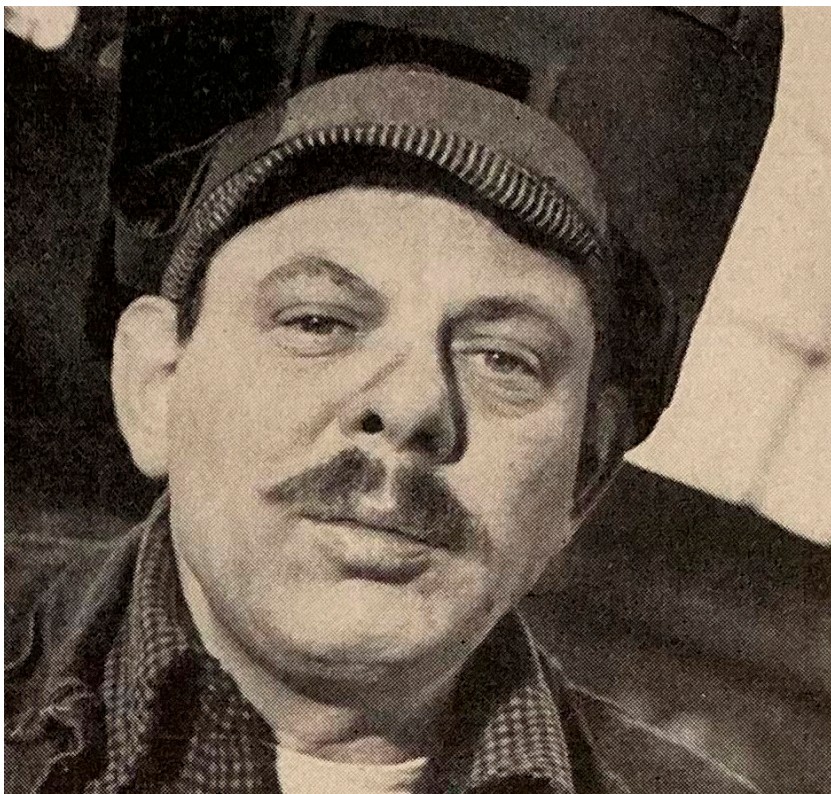

**Figure 2.** David Smith, 1952. Photograph by John Stewart.

## 2. Early Career

Although this was the first time Smith appeared in the pages of *Life* magazine, his manner of working metal to make sculpture had been the focus of publicity from the earliest stages of his career. In 1938, at the time of his first one-person exhibition, a "portrait at work on one of his forged steel sculptures" appeared in *Magazine of Art* (Figure 3).[4] With torch burning and sparks flying, this dynamic image had much in common with the opening *Life* photograph despite the fact that Smith employed an oxy-acetylene torch as opposed to the arc-welding method captured in 1951. Still, his protective gear—goggles, vest, and apron—virtually eliminated any conventional indicators of identity. A single image presented alone, as in this instance, replaced the tradition of the Romantic genius with one of an anonymous worker.

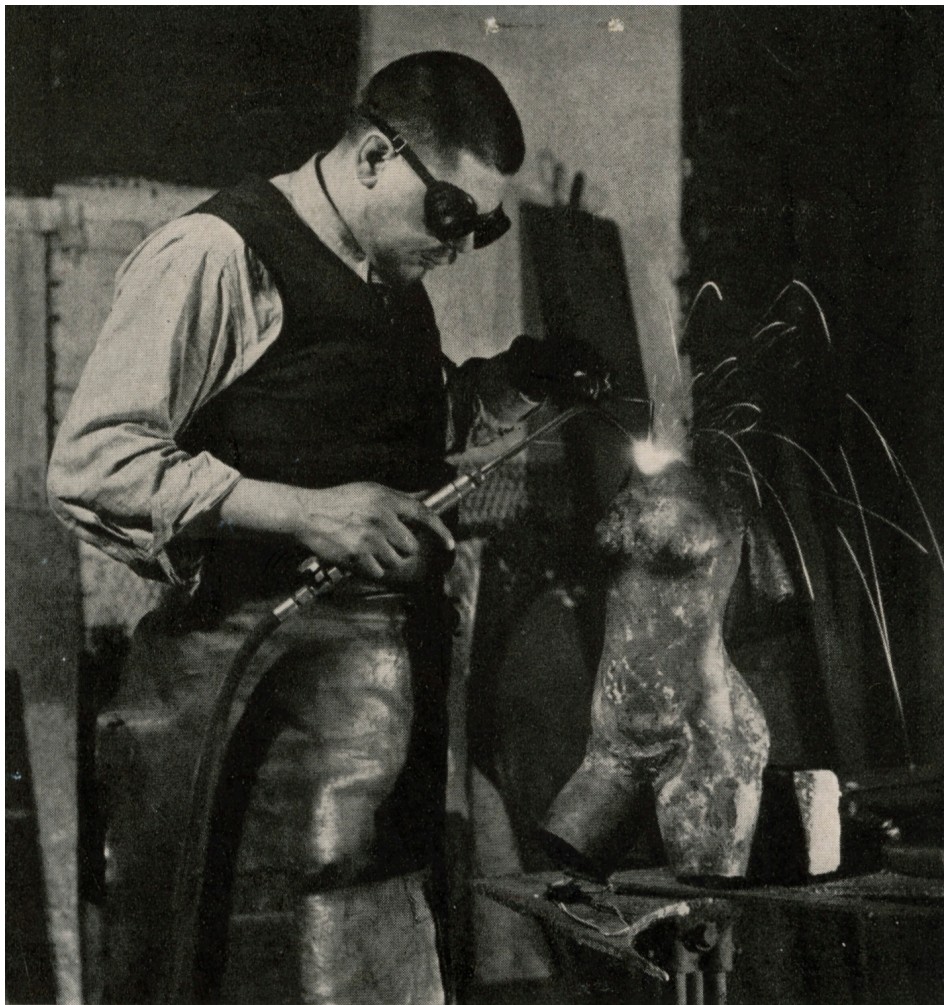

**Figure 3.** David Smith welding, 1938. Photograph by Leo Lances.

Beginning in 1933, Smith began producing the metal sculpture upon which his reputation would be based, but, for the next five years, there were few opportunities to show his work. The low trajectory of his career in the 1930s was typical of artists of his generation who would later be associated with the New York School.[5] Smith's status as an American, his commitment to modernism, and especially his choice of medium were three serious strikes against him in the New York art world of the pre-war era. Not surprisingly, given these circumstances, the first publications related to Smith's sculpture were brief, often without illustrations of any sort, and served as little more than announcements for his exhibitions.[6] However, a pattern quickly emerged of authors stressing the processes involved in working metal to make sculpture.

Over time, Smith would come to be celebrated as the first American artist to embrace welding.[7] However, in the 1930s, the art world's incomprehension of metalworking as a technical process contributed to the need for basic explanatory information. This unfamiliarity was never more evident than when Alfred Barr, curator at the Museum of Modern Art, first looked at Smith's sculpture and reportedly inquired "What's holding it together? Chewing gum?"[8] The methods Smith employed also seemed to require justification as legitimate for art making, especially since the images used to demonstrate his metalworking presented him as an industrial worker. One way to attempt to balance these potentially conflicting goals was to show Smith applying his novel manner of working in the production of more traditional objects. In fact, there was a marked tendency in early published images of Smith at work to show him sculpting the female figure, as was true in *Magazine of Art*, where he was welding *Steel Torso* (1936). Admittedly, the human form predominated

in Smith's early sculpture, and many of these objects were specifically gendered as female. However, the published examples were more conventional—both in level of representation and approach to form—than his other works from the same period. This fact was even acknowledged in the *Magazine of Art*, where the caption stated that many of the works in the announced exhibition were "more abstract than this".

The image of Smith sculpting a woman's body, rendered as solid and integral, brought a notable measure of tradition to photographs of him at work despite the framing notion of the art as new and different. A similarly conflicted message about the artist, his art, and his manner of working was apparent when three photographs of Smith in his studio appeared in a 1940 issue of *Popular Science.* The one-page article appeared in the table of contents under the category "Unusual Facts and Ideas, thus positioning the artist as a curiosity worthy of interest to the readers of a publication aimed at those "who wanted to know something about the world of science".[9] The headline assigned him a hybrid identity—"(Blacksmith-Sculptor Forges Art 1940) "—and the brief text called attention to his inventive finishes. In doing so, it called out details likely to appeal to the hobbyist, such as "Smith has developed a fireproof cherry-red color, which he mixes with hard was and melts upon the cooling metal".[10] However, in each of the accompanying photographs, the artist is shown shaping, polishing, or cutting a distinctly female form.

A similarly bifurcated message was conveyed by Maude Riley's profile of Smith published in 1940.[11] Her title identified Smith as a "Forward-Looking Modern Artist", yet the article was illustrated with a version of the same photograph that appeared in *Magazine of Art* two years earlier. Moreover, although Smith wore goggles and a leather apron and the photograph captured sparks flying from the acetylene torch, it seems likely that the photograph was staged given that he was wearing a collared shirt and tie.[12] This is a reminder that, to some extent, Smith was complicit in the construction of all his studio images, participating in their production and at times likely acting out his process for the photographer.[13] In this instance, the image of Smith at work was accompanied by a reproduction of *Growing Forms* (1939), a more radical, open-work sculpture. However, the caption described it as "inspired by the human form, which he draws and studies constantly", thus positioning the work in relationship to *Steel Torso*.

If there is a noteworthy connection between a technique new to high art and more traditional form in these early publications, there is a telling difference in the approach provided by Smith himself when, in 1940, he was given the opportunity to write an article on the role of sculpture in architecture for *Architectural Record* (Smith 1940). Technique continued to be the focus, but, in this instance, there was no equivocation about the new in sculpture. Instead, there was an unqualified emphasis on "unity between modern concepts, modern materials, and modern tools".[14] Smith and his sculpture were the subject of two of the article's illustrations: one was a reproduction of his *Vertical Structure* (1939) and the second showed Smith at work (Figure 4). Similar to the *Magazine of Art* photograph, this image showed him encased in protective gear in the process of welding. However, in this instance, the object under production was, as in *Vertical Structure*, an example of his more abstract sculpture.[15] Both works exploited welding's potential to shape metal into open forms rather than treating sculpture as solid and space-displacing. Thus, in the context of Smith's own presentation of contemporary sculpture, his innovative way of working was linked with radical sculptural forms. In contrast to other published reports of this period, Smith's account suggested that he was more willing to affirm the avant-garde aspects of both his technique and the resultant objects than were critics and publicists.

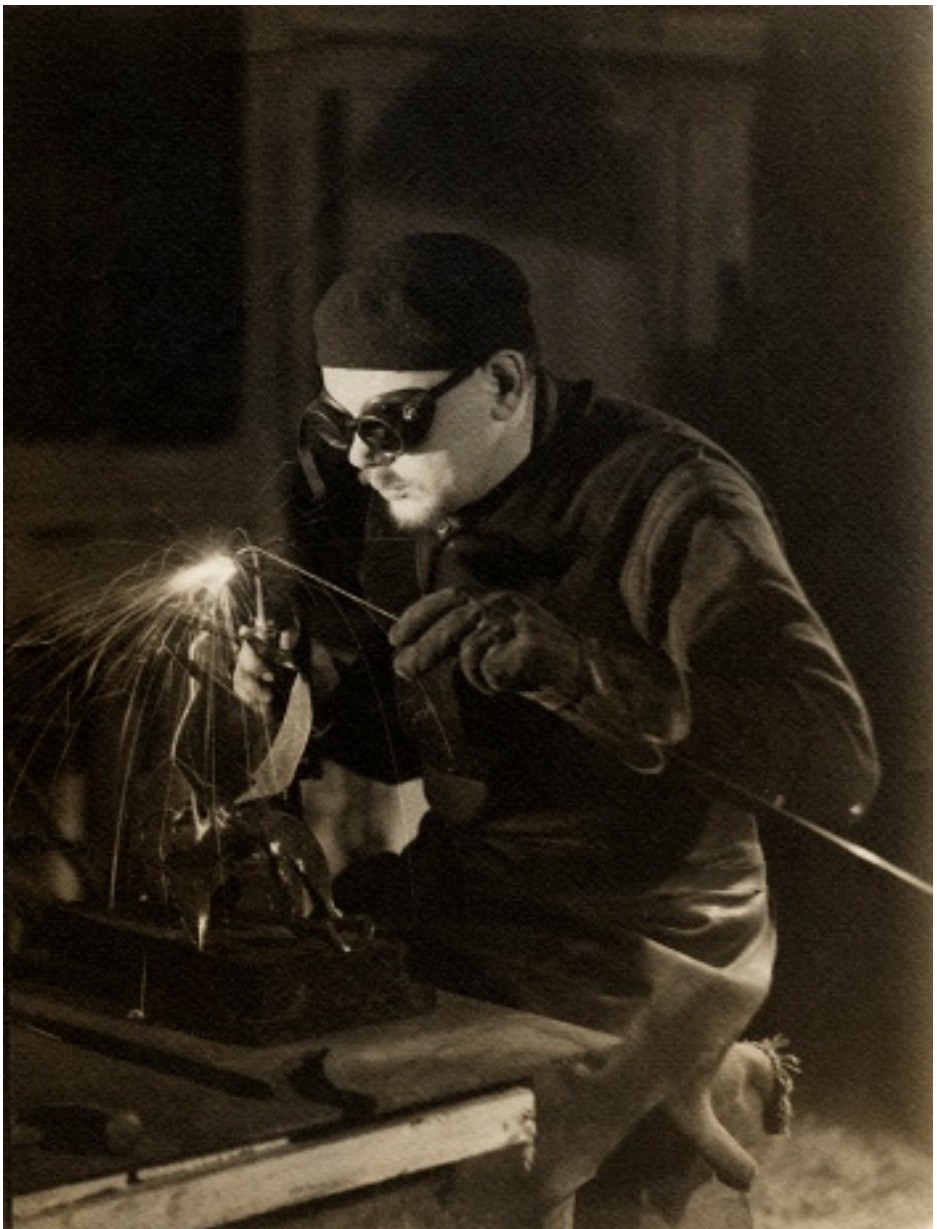

**Figure 4.** David Smith welding, 1940. Photograph by Robert McAfee.

There can be no question that Smith recognized the differences between his figural works and his more avant-garde sculptures and was aware not only of the former's association with tradition but their greater marketability. In 1940, he wrote to his dealer, Marian Willard, "If I stuck to torsos and sensual stuff I probably could be a success. I can sometimes dispose of a torso which I may have made years ago—in preference to new work".[16] At some level, his comment responded to the information captured in those early published photographs, where his mode of working is exploited as different, while his sculptural forms are relatively conventional. Nonetheless, those same female torsos, resulting in heterosexually oriented sensuality, serve as a stark reminder that Smith's modernism was, as with that of the European modernism that preceded it, built on images of women.[17] For Smith, the struggle was between what might bring him financial success and advancing his art. He persevered in making "new work".

### 3. World War II

In comparison to the limited press coverage Smith received in the late 1930s, there was a virtual explosion of notices about his career during World War II, with articles appearing in art periodicals as well as newspapers and even *Newsweek*. To a great extent, this development was attributable to the artist's burgeoning career as, although Smith had had only one solo exhibition in the late 1930s, in the early 1940s, he had several—in New York City and elsewhere.[18] However, in addition to the increase in sheer number of published items, there was a different tone to many of the articles of the war years: in place of the tendency to explain and defend his chosen technique, there was a new emphasis on the man himself. This stemmed primarily from interest in Smith's wartime employment as a commercial welder at the American Locomotive Company (ALCO) in Schenectady, New York, which provided an excellent journalistic hook for the period and was utilized again and again by those reporting on the artist's exhibitions. Thus, interest in one sort of work stimulated greater interest in another. Yet, always, it was Smith the artist who legitimized reporting on Smith the worker.

Virtually every published piece on Smith from this era made reference to his employment at ALCO, exploiting the link between his war work and his method of making sculpture, with headlines such as "Welder-Sculptor", or, less succinctly, "Sculptor, Alco Employe Will Exhibit Work: David Smith Uses Metal Knowledge as Welder" (Welder-Sculptor 1943; Bradt 1943). In keeping with this emphasis on the man and his activities, the articles were illustrated, in almost every case, with photographs of Smith rather than reproductions of his sculpture. However, while these photographs regularly presented Smith making art, there was a marked tendency for him to be shown doing something other than welding. Instead, he appeared using more traditional tools, or merely posing with his sculpture—something partly explained by the fact that most of these photographs were taken in the temporary living quarters Smith occupied in Schenectady while he was employed at ALCO.[19] Smith's workspace, as well as the methods he could employ to make art, were severely restricted in these cramped rooms, as is evident in the photograph that illustrated the 1943 *Newsweek* article (Figure 5).[20] In this image, Smith stood in his shirt sleeves at the small table that served as his studio space in the Schenectady apartment, drawings and clippings mounted on the wall beside him. Looking every bit the traditional sculptor, he wielded a hammer and chisel while at work on *Sewing Machine* (1943). This Danby blue marble form was one of a handful of stone carvings he produced during the period, a move prompted by wartime shortages of metal (Lyon et al. 2021). Although some finishing work may well have been conducted in the rented rooms in Schenectady, the carving for these stone pieces was primarily accomplished with the aid of an air compressor at the monument works Mallery and LaBrake in nearby Saratoga Springs.[21] However, the photograph did nothing to inform the viewer that, with stone, as with metal, Smith had adapted an industrial carving technique to sculpture.

The stark contrast between the manner in which the photographs in these wartime publications presented Smith and the information provided by accompanying text reached dramatic heights in a 1943 article by Maude Riley, now writing for *Art Digest* (Riley 1943). In line with other reports from the period, Riley's words told of a war worker who also welded and cut metal to form sculpture. She provided a picturesque written description of the artist as weighing "220 pounds" and looking "like Wallace Beery when he gets shop-mussed". Yet, the photographic illustration to her article shows a well-groomed and thoroughly respectable figure, inspecting one of his metal sculptures while standing at the Schenectady worktable (Figure 6).[22] No effort was made to acknowledge the dichotomous aspects of this presentation; rather, such a clear distinction between the twin poles of Smith's endeavors seemed taken for granted.

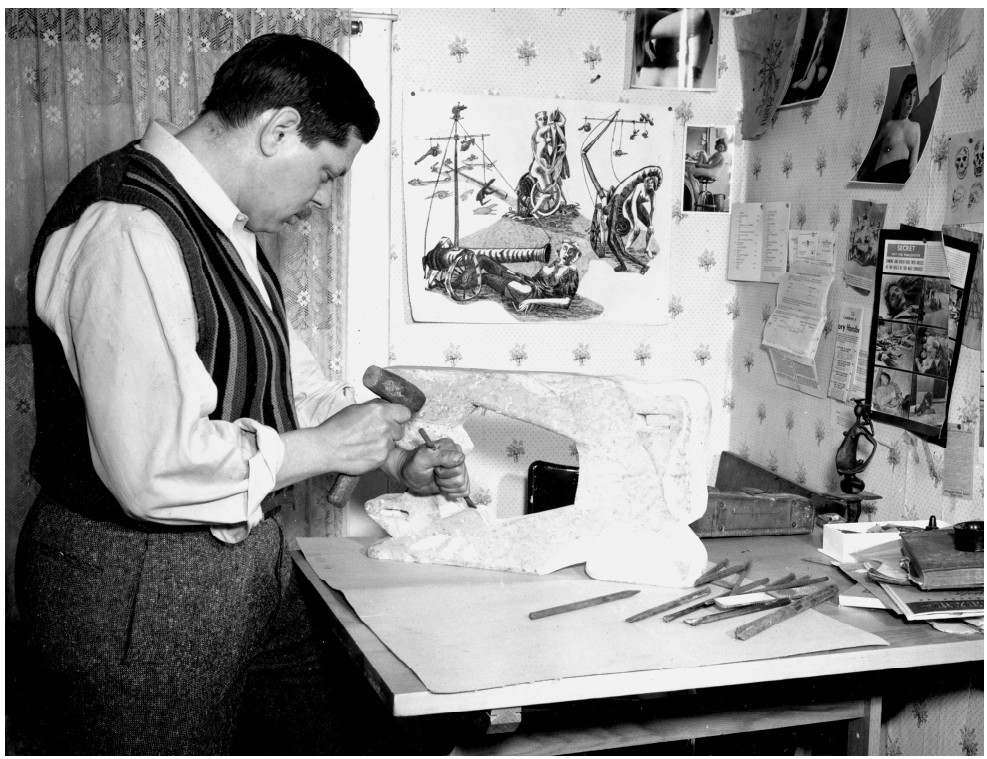

**Figure 5.** David Smith carving Sewing Machine, 1943, in his Schenectady apartment, c. 1943. Photographer unknown.

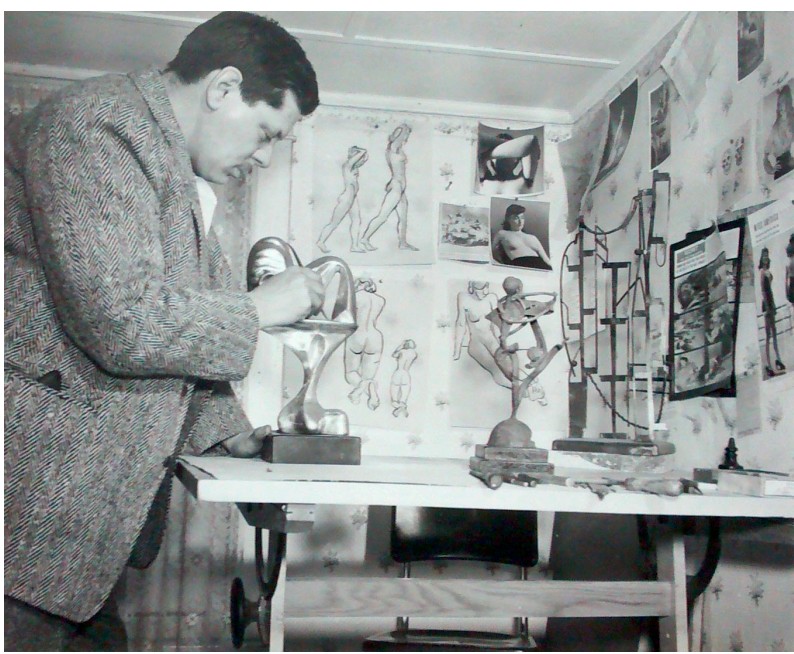

**Figure 6.** David Smith in his Schenectady apartment, c. 1943. Photographer unknown.

Admittedly, the circumstances of Smith's wartime life curtailed the options under which photographs could be taken. The end result was that he was shown engaged in more conservative art-making techniques, wearing more conservative clothing, and located in more traditional settings than in published images from the late 1930s. Earlier photographs had informed the public as to Smith's use of welding as a means to produce sculpture and served to legitimize this method in the realm of art (if, as has been shown, they regularly linked him to more conservative sculptural forms). However, when attention was brought

to his wartime employment "as a skilled armor plate welder on M7 tanks for the U.S. Army", photographs used more traditional codes to present the artist, perhaps avoiding any erosion of the boundary between artist and worker.[23]

That Smith was not shown at the ALCO plant in these articles only served to strengthen the more conventional aspects of the published photographs from the early 1940s. Therefore, while the text of accompanying articles stressed the twofold nature of his life during this period—artist and worker—the visual images reinforced the notion of Smith as a contemplative, creative figure. ALCO, on the other hand, was aware its employee's other life as an artist and sought to exploit the potential for publicity inherent in Smith's dual role. The company arranged for a series of photographs to be taken "on tanks and at home", and a press release was prepared by Earl Newsom and Co. of New York City. However, that version of the Smith story appears to have gained little traction and did not penetrate the art world.[24] Instead, where the journalistic focus on his employment at ALCO might have shifted attention toward his role as a worker, the photographs that were published served to redress the balance in order to emphasize "art" at the expense of "work". Thus, at a moment when Smith's role as a worker was before the public in its most unadulterated manner, some of the potential impact of that fact was mitigated by photographic images that presented him as a traditional artist. As a result, a clear boundary was maintained between what he did as a worker and as an artist.

## 4. Post-War Years

In the late 1940s, several years passed without the publication of another photograph of Smith at work. In fact, with very few exceptions, a gap of some seven or eight years exists between the photographs of 1943 and the next important group of images, which appeared in 1951.[25] The virtual absence of published photographs of Smith at work during this interval is in no way indicative of a dormant period in his career. On the contrary, during these years, he continued to hold regular one-person exhibitions at the Willard Gallery—in 1947 and 1950—in addition to a joint exhibition Willard organized with the Buchholz Gallery in 1946. He participated in a number of significant group exhibitions, including traveling exhibitions organized by the Detroit Institute of Arts and The Art Institute of Chicago.[26] Publications from this period were primarily tied to these exhibitions—announcements and reviews—and were illustrated almost exclusively with reproductions of the artist's sculpture.[27]

There is no obvious way to account for the disappearance of a type of photograph that had appeared regularly in the earlier years of Smith's career. Perhaps the explanatory function served by those early photographs was no longer considered necessary at a time when welding had become more widely accepted as a legitimate form of art making. Further, with Smith's ALCO employment ended, possible confusion about his status as artist or worker was no longer directly before the public. However, the emphasis previously placed on Smith's manner of art making was acknowledged as early as 1946 when one commentator noted "that until 1943 writers felt that they had to give much space to descriptions of [his] process . . . ".[28]

Still, by 1951, there was renewed emphasis on Smith's process of making art, as evidenced by two notable articles: *Life* magazine "A Sculptor Forges Iron", from September 1952 (discussed above) and "David Smith Makes a Sculpture", which appeared in *Art News* in November 1951. This pair of articles reflected an upsurge of interest in Smith's career, which paralleled developments in the lives of other artists of his generation. Indeed, these two articles were so representative of contemporary publicity surrounding other American avant-garde artists that they offer an important opportunity to consider differences between these subjects as treated by popular magazines as opposed to mass art magazines.

As was already mentioned, Smith's career had garnered greater interest than ever before in the years following World War II. This trend continued and intensified in the new decade as Smith continued to hold annual one-person exhibitions at the Willard Gallery. In 1953, his work was included in *Twelve Modern American Painters and Sculptors*, organized

by the International Council of the Museum of Modern Art and circulated to European museums, and, the next year, a retrospective of his work organized by the Cincinnati Modern Art Society travelled to several Midwest university museums. Concomitant to these exhibition opportunities was greater critical praise for Smith. Clement Greenberg, who was solidifying his role as a leading figure in modernist criticism, had been an early champion of Smith, and, by 1952, called him "possibly the most powerful yet subtle sculptor ... this country has yet produced, certainly the best since Gaston Lachaise".[29] This sort of acclaim combined with the publicity surrounding him and his work helped to solidify his reputation as the most important American sculptor of his generation.

Of course, the blossoming of Smith's career in the early 1950s paralleled developments in the lives of other artists of his generation. These artists of the so-called New York School came to prominence around 1950 as the result of a complicated set of circumstances that has been the subject of careful study.[30] Even to summarize the many factors involved is beyond the scope of this project; however, among the most important were a more positive climate for American avant-garde artists, arguably political in its motivation, and the favorable post-war economic conditions in the United States. These circumstances, combined with the individual achievements of the various artists involved, culminated in an unprecedented "triumph" for American art.[31] At the same time, the celebration of these artists—accompanied by photographs of them at work—called attention to the entire process of making art, which complemented a particular understanding of modernism.

Both popular magazines and mass art magazines embraced this increased interest in images of the artist at work. However, the message constructed from the images differed from one to the other. Popular magazines emphasized innovation and in doing so relied on these photographs to communicate that content, while mass art magazines were more concerned with balancing such photographs with information that affirmed connections to tradition. Generally, the message offered by mass art magazines was that recent developments in artistic practices may well have transcended the past but also had solid roots in the history of art. Such specialized magazines had little call to sensationalize the avant-garde, as occurred in publications catering to a broader audience. For mass art magazines, establishing links to tradition was an effective strategy that served to legitimize the work of the Abstract Expressionist artist for their audience of readers already interested in the art world. It might seem logical that a similar approach of presenting radical ideas couched in terms of the familiar would serve popular magazines equally, if not more, successfully. However, in fact, popular magazines, with the reputation for keeping readers up to date with the latest developments in a great variety of fields, had more to gain by stressing the startling newness of these artists' working methods.

Although there has been some debate over the extent to which popular magazines were actually supportive of the Abstract Expressionists, there is no question that they provided a good deal of coverage of these artists.[32] With attention focused on the artist at work, photographs offered a glimpse of the creative process and intimate insights into fascinating personalities. *Life* magazine's 1952 pairing of the dramatic full-page photograph of Smith welding (Figure 1) with the "David" image where the camera peered up into his face (Figure 2) was fully in keeping with this approach.

Mass art magazines, on the other hand, presented the phenomenon of the modernist artist as part of a continuum of art making, not as existing in an exotic, separate sphere of life to be gawked at as in a side show. The image of the artist at work was a mainstay of such publications in the 1950s but was presented in a more reasoned manner than the approach found in popular magazines. In fact, Mary Bergstein has pointed out the "close readings of the photographs and texts [...] disclose that traditional narrative codes supported modernist interpretations".[33] Nowhere was this positivist approach more evident than in the series of articles "X makes a work of art" that appeared in *Art News* during the 1950s. Each contribution to this series focused on the execution of a single work. The text of each article relied heavily on interviews with the artist. Progress toward the completion of the work was documented by a group of photographs presented sequentially. Attention

was balanced between images of the artist at work on the piece and reproductions of the work itself. As a result, the process of art making recorded in the series was not a single representative act that stood for the creative moment. Instead, one saw a series of steps, often including preparatory stages and/or a variety of methods employed to achieve the finish product.[34]

Smith's appearance in the series closely followed this format.[35] The article documented the making of *Cathedral* (1951) and opened with a photograph of the artist standing at a worktable in his studio drawing on a sheet of steel with soapstone (Figure 7). Clothed in jeans and denim jacket, cigarette dangling from his mouth, he was very much the brooding rebel of the age. This trope was familiar to the popular audience, not only because of the Romantic image of the tortured Bohemian artist before the public for almost a century but thanks also to the emerging image of the brooding rebel in post-war American society as exemplified by film performances by Marlon Brando and James Dean, and in literary circles by the Beats.[36]

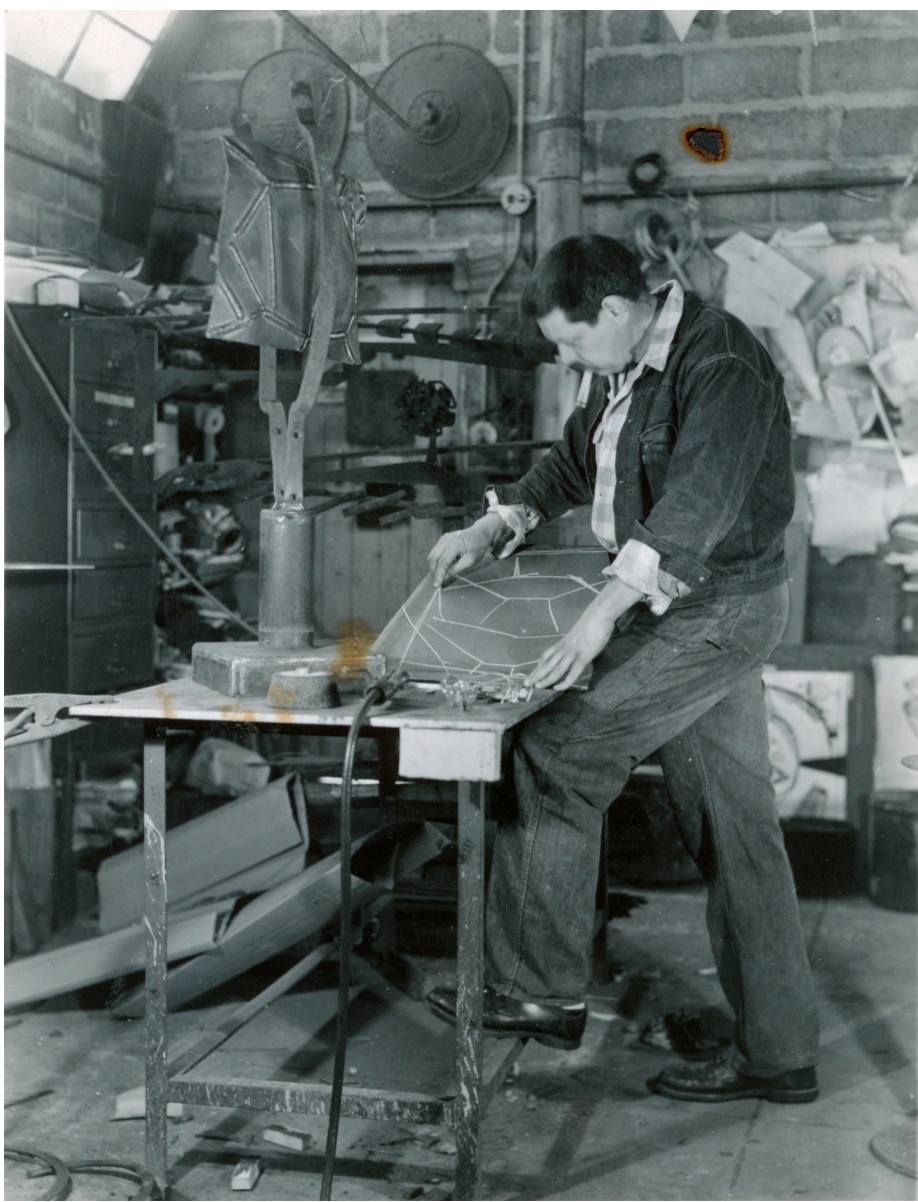

**Figure 7.** David Smith working on *Canopic Head*, 1951, in his Bolton Landing workshop, c. 1951. Photographer unknown.

Elaine de Kooning, the article's author, reinforced this visual image with her text as she presented Smith as an artist–hero who lived a solitary existence while struggling to make art through an intimate, meaningful exploration of his materials. As Lee Hall pointed out, despite the reliance on carefully recorded details specific to Smith's life, the definition of the artist as presented in de Kooning's article could and would be applied to any and all of the Abstract Expressionist artists as reality mixed with myth in forging a group of post-war heroes in the 1950s (Hall 1993).

However, at the same time, in keeping with the established format of the series, the article included several images of Smith's *Cathedral* in various stages of completion along with five additional photographs of him at work. These images captured him executing a range of exacting technical skills—forging, sawing, cutting, brazing, and welding. The equipment required to carry out each of these processes became a prominent part of each image. Thus, the artist was subsumed into his working method, while the emphasis remained on his technical expertise carrying out these procedures. For the *Art News* series, this was the approach to process—not drama but a considered series of actions.

## 5. Final Years

In February 1960, *Arts* magazine published a "Special David Smith Number" with a lengthy article by Hilton Kramer, then editor of the magazine. The issue appeared the same month a large solo exhibition of Smith's recent work opened at French and Company, and together these events signaled a new level of recognition in which even the sometimes cantankerous Smith could take pride.[37] Kramer's enthusiasm was evident from his opening sentence in which he offered unequivocal praise for Smith's sculpture as "one of the most significant achievements of American art, not only at the present moment but in its entire history" (Kramer 1960). The article went on to provide a detailed critical assessment of Smith's accomplishments while delivering a thoughtful overview of his career. The text was supported by more than three dozen reproductions of Smith's sculptures, presented in chronological groupings. In addition, there were nine portrait photographs of Smith. Of these, three showed him in the process of working with metal. These images had been published elsewhere and included the "first" photograph of Smith welding *Steel Torso* (Figure 3) and the opening portrait from "David Smith Makes a Sculpture" (Figure 7). The summary nature of the entire project was underscored by the fact that all photographs were attributed to Smith, even though some of them previously had appeared in print with the credit line of professional photographers.[38] All in all, the images—and Kramer's text—emphasized the consistency of Smith's enterprise as an artist rather than accentuating the newness of his work in the post-war years.

If the impact of Kramer's article and its accompanying illustrations was to acknowledge the continuity of Smith's career, in fact, the most influential corpus of photographs of Smith at work was yet to come. In the final years of the sculptor's life, from 1962 to 1965, the sheer number of photographs in circulation increased and he was associated with photographers of significant reputation—notably Ugo Mulas and Dan Budnik. Both photographers were closely tied to the New York art world of the 1960s, although neither made it his exclusive subject. Without question, interest in their photographs of this milieu was an outgrowth of the attention focused on the personality of the artist and the process of making art by photographs published in popular and mass art magazines in the 1950s.[39] Each had the opportunity to photograph Smith at work during a sustained session—Mulas in Voltri, Italy and Budnik at Smith's home in Bolton Landing, New York. These photographic campaigns and the resulting publications helped to construct and disseminate Smith's legend through an exchange between artist and photographer.[40]

Mulas first encountered Smith in June 1962 when he was working in Voltri, Italy. Smith had come to Italy at the invitation of the Spoleto Festival of Two Worlds with the expectation he would produce one or two works during a month-long residency.[41] The project was sponsored by Italsider, the Italian national steel company, which provided Smith with workspace in abandoned factories, access to old machine parts, tools, and

steel remnants, along with the assistance of several metalworkers. These circumstances were close to ideal for Smith and resulted in a remarkably productive few weeks, during which he produced twenty-six large steel sculptures. Giovanni Caradente, the festival's art director who made the original arrangements with Smith, immediately positioned the sculptures in Spoleto's Roman amphitheater as well as throughout the town.[42]

In 1964, a selection of Mulas's photographs appeared in *Voltron*, a slim, well-designed book published by the University of Pennsylvania.[43] Mulas' images chosen for this publication were almost evenly divided between portraits of Smith and works in progress in the factory at Voltri and completed sculpture on display in Spoleto. Compared to other photographers, Mulas placed much less emphasis on the dynamic physical process of making sculpture. Although two of the published photographs showed Smith wearing a welder's helmet, none showed him in the process of welding—or doing any other type of metal working. No doubt partly in response to the enormous spaces of the factory, several of the photographs were composed so that Smith occupied the middle ground, making him less of a commanding presence in the spaces he occupied (Figure 8). These tendencies in Mulas' photographs resulted in a relatively subdued and thoughtful image of the artist, quite different from dynamic images that frequently captured the attention of other photographers.

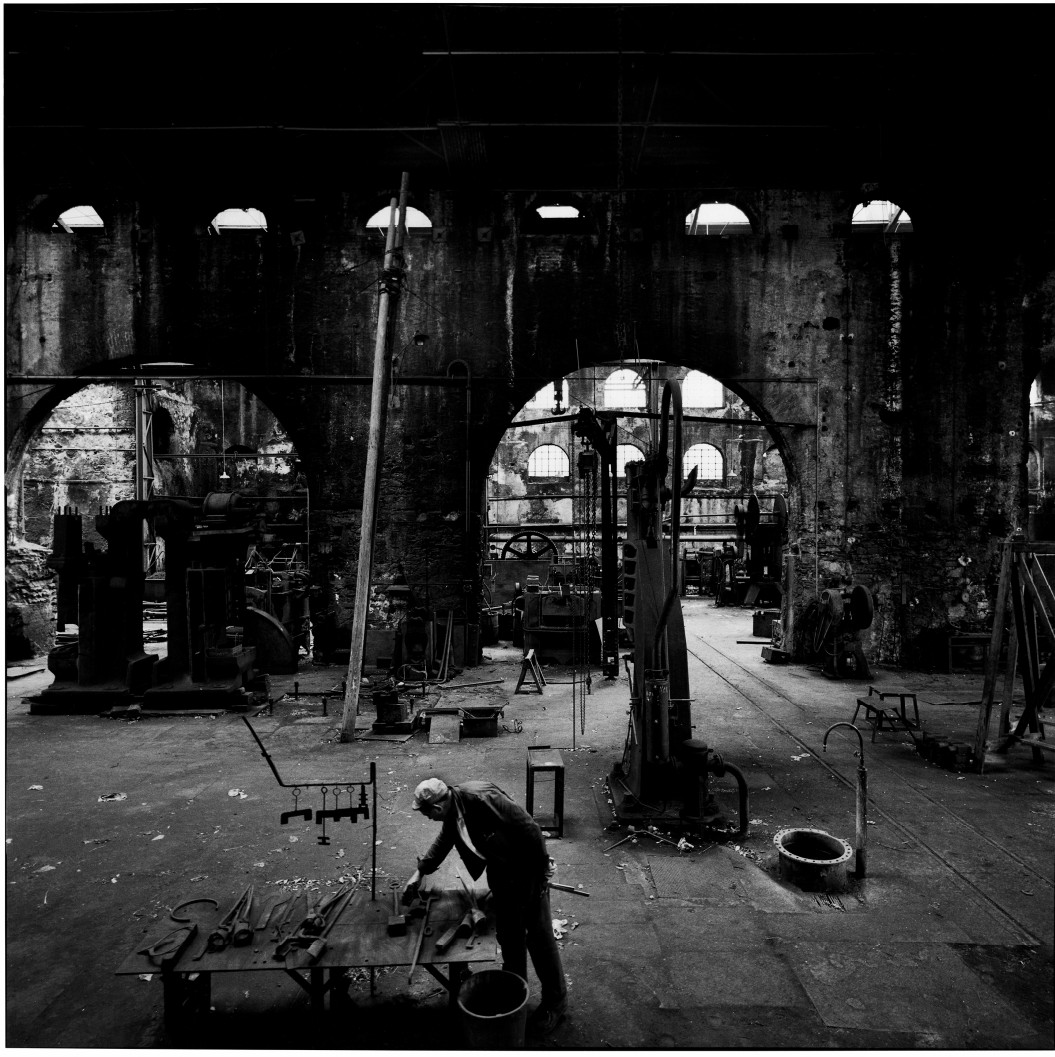

**Figure 8.** David Smith at work in the Italsider factory, Voltri, Italy, summer 1962. Photo Ugo Mulas © Ugo Mulas Heirs. All rights reserved.

*Voltron* also included Budnik's photographs of the *Voltri-Boltons*, a series of sculptures Smith had fashioned after returning to Bolton Landing using materials from Voltri that Italsider had shipped to him at his request. However, Budnik's identification with Smith had already been cemented during a visit to Bolton Landing in December 1962. A selection of his photographs of Smith at work appeared in *Life* magazine the following Spring (David's Steel Goliaths 1963).

In this second appearance on the pages of *Life*, Smith was no longer the novelty to be defended as he had been in the early 1950s. Along with other artists of his generation, he now enjoyed an established reputation—a matter reflected, as pointed out by the article, in the current asking prices for his sculpture "from $5000 to $50,000".[44] This success, combined with the public's continuing appetite for images of contemporary artists, merited five full pages in the magazine. Although the photographs were more numerous than those in the first *Life* article devoted to Smith, they still conceptualized the artist in a similar manner, combining images of a man of action with one in contemplation. Thus, they attempted to convey both the hard work and intellectual effort responsible for Smith's sculpture.

The interest in showing Smith as a metal worker remained, and one of the largest photographs showed him welding (Figure 9). Reflecting a shift to working on the floor that had taken place in the early 1950s, Smith was kneeling and turned away from the viewer. At that angle, the bulk of his body blotted out the site of the weld but did not obscure all of its intense light. As a result, the photograph was an image of industrial production at the same time that it managed to convey a sense of mystery about the process. What was unseen stood in for the creative genius at work. Still, in the context of the entire layout, this and the other action photographs reinforced the physical exertion of his labor. In a smaller image, Smith was captured with legs firmly planted and hammer raised as he prepared to strike metal against an anvil. The remaining two action photographs spoke to the sheer effort required as Smith grappled with his larger-than-life sculptures, a theme verbalized in the article's headline "David's Steel Goliaths" and which positioned Smith as larger than life, a fitting Abstract Expressionist hero (Figure 10).

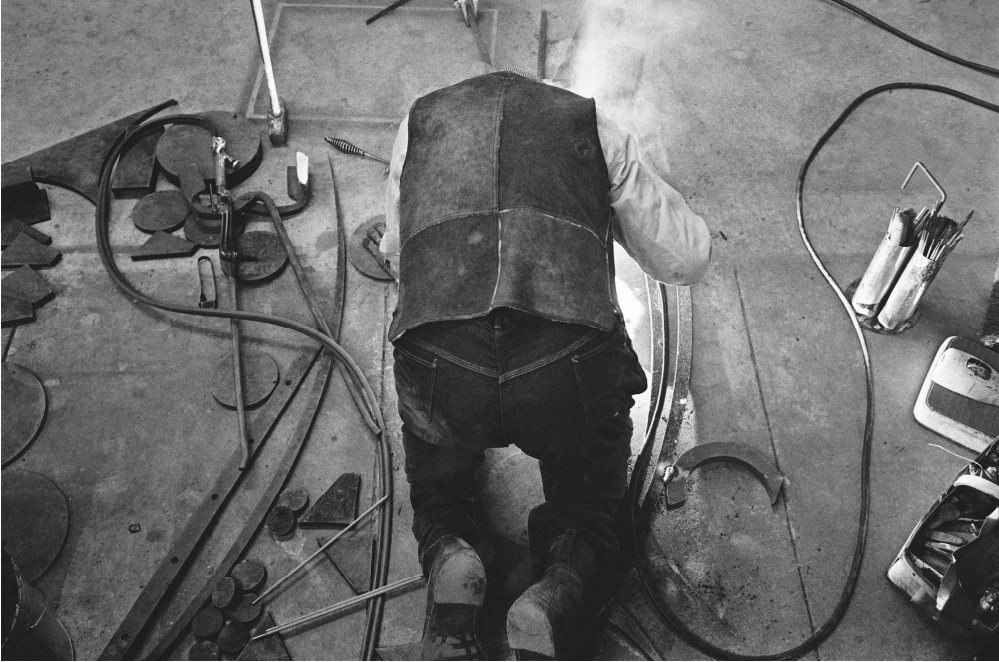

**Figure 9.** David Smith at work, Terminal Iron Works, Bolton Landing, New York, 1962. Photograph by Dan Budnik. © 2023 The Estate of Dan Budnik. All Rights Reserved.

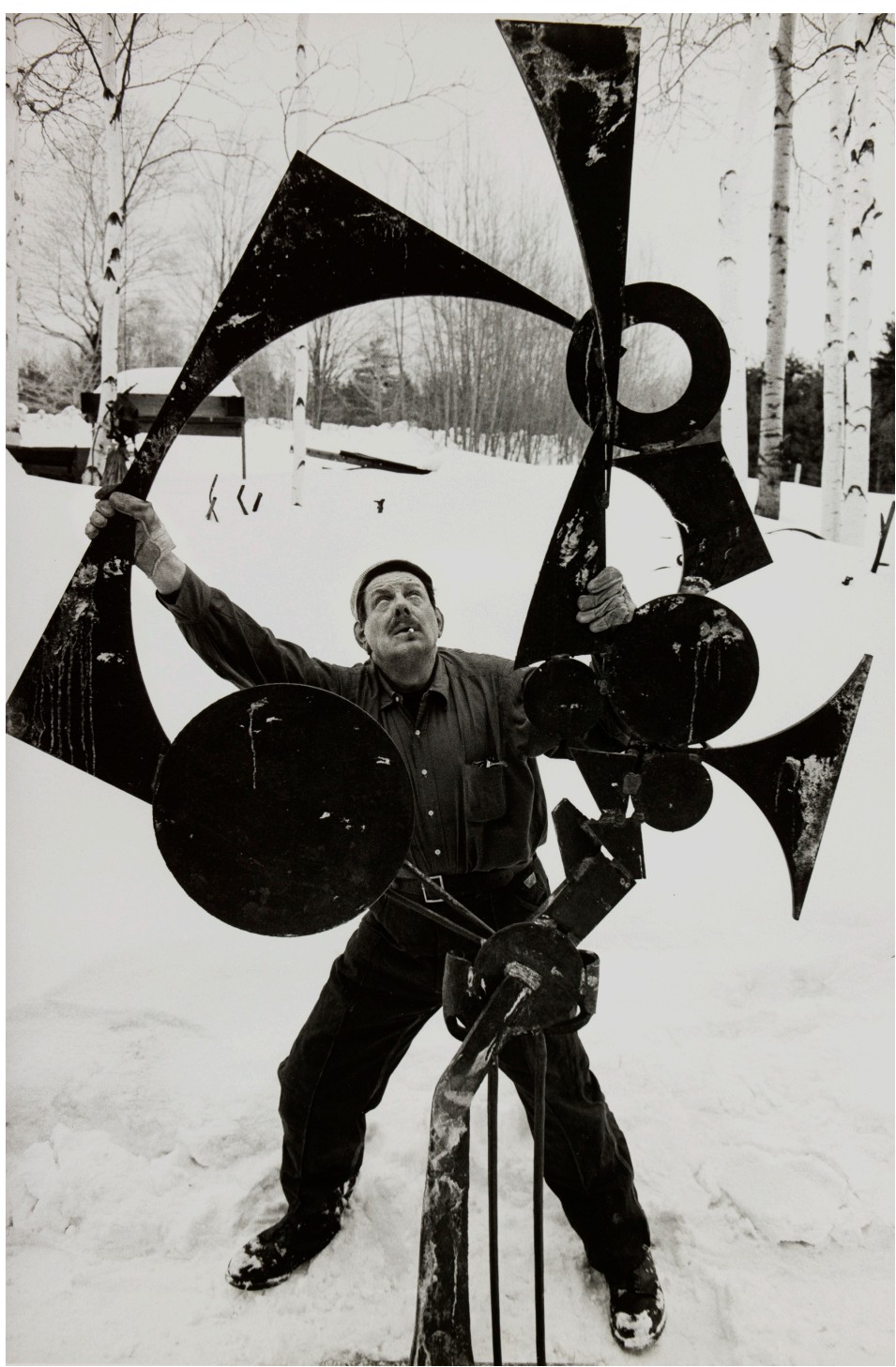

**Figure 10.** David Smith with *Voltron XVIII*, Terminal Iron Works, Bolton Landing, New York, March 1963. Photograph by Dan Budnik.

As had happened previously, there was a disjunction between the information provided in the article's text and the message of its photographs. While the article referenced Smith's use of assistants, the photographs showed him undertaking demanding physical tasks alone. In reality, after World War II, Smith was increasingly able to afford to hire people to assist him in Bolton Landing and took advantage of the possibility. For example, in 1962—the year Budnik visited Smith's workspace—records indicate that there were three people working for him.[45] The most important of these was Leon Pratt, a neighbor who worked for Smith beginning in the late 1940s, at first on a part-time and later a full-time

basis. Smith's assistants at Bolton Landing facilitated the increased scale of sculpture and the elevated rate of production in the later years of his career. Therefore, when Italsider provided him with six local factory workers to serve as his assistants in Voltri, Smith was comfortable with this manner of working. Without question, at Bolton and at Voltri, his use of assistants echoed the various periods in his life—including at ALCO—where he had been employed as a commercial metal worker working side by side with others. There is every indication that Smith took satisfaction in the comradeship experienced with his assistants; nonetheless, when he functioned as an artist, there was a clear hierarchy in place between him and those who labored at his direction.[46] This reality was further underscored by the absence of those assistants in published photographs. In their place, attention was focused on Smith's difficult, challenging work and apparently solitary accomplishments.

If the emphasis on Smith's physical labor fed the interest in the process of making art, it simultaneously fit the era's concept of the macho artist. Big, strong, and capable, his physical size and strength were regularly noted in profiles; in 1964, *Life* magazine described him as a "monolithic man" who worked "steadily and wordlessly".[47] The level at which masculinity became intertwined with the meanings assigned to post-war art has been closely studied, primarily in relationship to Jackson Pollock, who often serves as the representative Abstract Expressionist artist. According to Amelia Jones' important analysis of the topic: "Pollock aligned himself with recognizable codes of masculinity (hence of artistic authority) active in US culture at the time" (Jones 1995). Certainly, Smith's physique was that of a man's man, larger than average (and far beyond that of Pollock). In this context, physical type was literally bound up with clothing in the "formation of artistic identity", and clothing also served as a signifier of manliness for American avant-garde artists in the middle of the twentieth century (Nead 1995). Again, Pollock meaningfully represented this trope without having any monopoly on the practice. According to Jones, Pollock's decision to be photographed wearing jeans not only reflected practical attire he wore in the studio but—according to Jones—exaggerated "his affiliation with working-class masculinity when performing himself as an artist for the camera...".[48] Smith's clothes were treated in a similar fashion. For example, de Kooning likened Smith's winter work clothes to "those sported by lumbermen in the district" as she described his studio as only heated by two oil stoves despite frigid temperatures.[49] This sort of information elevated Smith above the ordinary through his association with robust craftsmen, thus foregrounding his own hardy physique and rugged individuality. The implication that Smith was fully prepared to challenge the elements embedded in her description also reinforced the notion of an extraordinary man—once again feeding the myth of the heroic artist.

Budnik's most famous photograph of Smith similarly conveyed a message about the man's imperviousness to the weather and—ultimately—the extent of his creative genius. As originally published in the 1963 *Life* magazine article, it was spread over one-and-three-quarters pages (Figure 11).

Seated out-of-doors with his back to the viewer, Smith was coatless despite the snow blanketing the landscape. Beyond him, the viewer saw the south field of his Bolton Landing farm planted with his crop of sculpture. Because the viewer understood the brooding presence seated on the bench produced the sculpture seen in the field beyond, he served not as a traditional *repoussoir* figure, mediating between the viewer and the landscape, but rather as a barrier separating us from it because we do not know—cannot know—the sculpture as he does. Here, any notion of the laborer was completely set aside, replaced by the artist as a force of nature with god-like creative abilities.

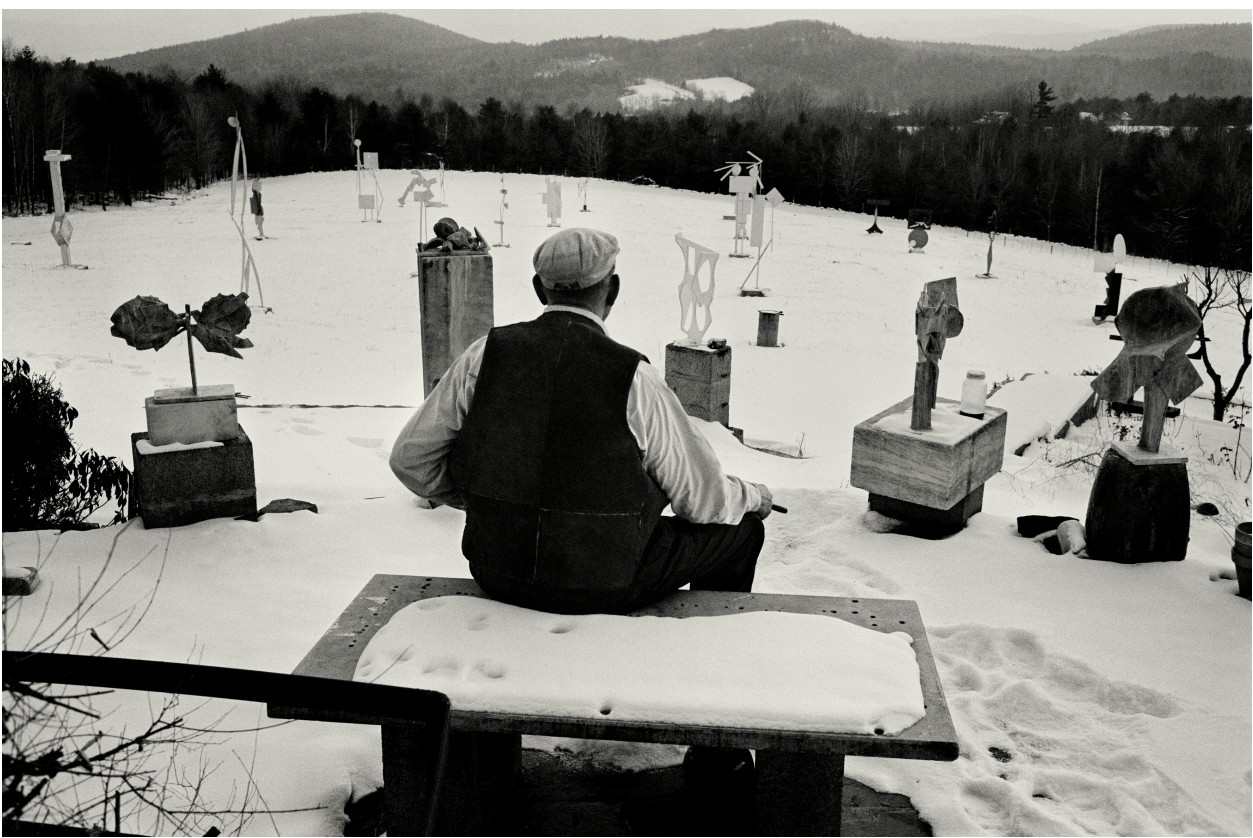

**Figure 11.** David Smith, South Field, Terminal Iron Works, Bolton Landing, New York, 1962. Photograph by Dan Budnik.

In May 1965, Smith was killed in an automobile accident (David Smith, 59, Metal Sculptor 1965). As a result, the sense of distance established by Budnik's photograph—a perception of the artist being in a place where the viewer is unable to go—gained even greater resonance. Thus, it was not surprising to find that the same photograph concluded *Art in America*'s tribute to Smith, published shortly after his death.[50] Smith's apotheosis was complete when, in its obituary, *Life* celebrated Smith as a pagan god with the words "Vulcan of American Art".[51] Of course, Smith's sudden, violent death, as with those of Arshile Gorky and Pollock, and later Mark Rothko, was "woven into the larger myth of the *artiste maudit*, the Promethean seeker punished by the gods" (Peppiatt and Bellony-Rewald 1982). In death, Smith was elevated to the level of his fellow painters and joined them on Mount Olympus. If that process had begun years before, it was not rooted in the beginning of Smith's career. As has been shown, early photographs were used to establish the legitimacy of his industrial methods. Only around 1950 did that information intersect with interest in the working process of his painter colleagues.[52] After all, photographs of Pollock painting only captured the public's attention once the information they provided served as a surrogate for the finished work.[53] Unlike those for Smith, there were no earlier published images of Pollock painting. However, once that model was established, Smith could fit the new mold. Nowhere was that more apparent than when he appeared in the *Art News* "X makes a work of art" series just four months after Pollock (Goodnough 1951). A broader celebration of art making may have solidified interest in Smith's practice, but, by that point in his career, it was carefully oriented toward his individual, unique status as a genius creator. Welding became alchemy.

**Funding:** This research received no external funding.

**Institutional Review Board Statement:** Not applicable.

**Informed Consent Statement:** Not applicable.

**Data Availability Statement:** The date used in this study are available in published sources.

**Acknowledgments:** This article originated with the paper "More than a Man, Less than a Painter, David Smith in the Pages of *Life*", presented at the Twentieth Century Session at the Midwest Art History Society Conference on 29 March 1996. In 1998, I submitted a much-expanded treatment to the *Art Bulletin* as "The Artist as Worker: Photographic Images of David Smith". Although I never completed the requested revisions, I reworked the ideas for the invited talk "Defining the Modernist Artist: Jackson Pollock and David Smith", delivered at Northern Illinois University, October 18, 2000. I welcome the opportunity to revisit the material here after a long hiatus, acknowledging that, in the meantime, related material was explored by Megan Bissonnette, "The Making of an American Sculptor: David Smith Criticism, 1938–1971", (PhD diss., York University, Toronto, 2014).

**Conflicts of Interest:** The author declares no conflict of interest.

## Notes

[1]   For more information on Smith's self-presentation and his political engagement, see (Wisotzki 2005).

[2]   The link between Smith's family name and aspects of his craft proved irresistible to numerous writers and critics. As in this instance, the reference was to a blacksmith—a craftsman who traditionally forged iron (black serving as a reference to the metal's color). Although the *Life* magazine photograph showed Smith welding steel, the magazine further underscored the ties to craft by labeling the article "A Sculptor Forges Iron" in the issue's table of contents, p. 25.

[3]   "Sculptor Forges", p. 76.

[4]   (Seeing the Shows: Exhibition in New York and Chicago Reviewed 1938) The exhibition was organized by Marian Willard and held at her East River Gallery, New York from 19 January to 5 February 1938.

[5]   During the years of the Great Depression, Smith shared with his colleagues the limited opportunity to exhibit art, and the distractions of a "day job" (in his case, technical advisor for the Treasury Section of Painting and Sculpture, Temporary Emergency Relief Administration).

[6]   Representative citations without illustrations and no longer than two paragraphs from 1938 to 1940: (Davidson 1938; Bird 1938; Lechay 1940; Around the Galleries: Five New Shows 1940).

[7]   Smith was influenced by the metal sculptures produced by Pablo Picasso and Julio Gonzales beginning in 1928. John Graham showed Smith reproductions of those sculptures in issues of *Cahiers d'art* (Brenson 2022).

[8]   Dehner interview with author, June 1985. Marcus, citing Dehner, attributes this observation to dealer Julien Levy (Marcus 1983).

[9]   *Popular Science*, July 1940, p. 4 (Nourie and Nourie 1990).

[10]   *Popular Science*, July 1940, p. 69.

[11]   (Riley 1940) *Cue* was a listing guide for New York City entertainments that included commentaries, such as Riley's article on Smith. See, (Mort Glankoff Dead and Began Cue Magazine 1986). D18. Riley remained an active art critic throughout the 1940s (Riley 1983).

[12]   Anne Wagner comments on Smith shown welding "wearing a sweater and tie". "David Smith: Heavy Metal", (*David Smith: Cubes and Anarchy* 2011).

[13]   Staging studio photographs was a practical and common occurrence. For Hans Namuth's account of working with Pollock, see his (Rose 1980b), n.p.

[14]   Smith, "Art Forms", p. 77.

[15]   The sculpture in process may be *Bathers* (1940).

[16]   AAA 986-782, David Smith to Marian Willard, no date (1940). (AAA citations refer to collections on deposit at the Archives of American Art, Smithsonian Institution).

[17]   For an important early statement on this topic, see Carol Duncan, "Virility and Domination in Early Twentieth-Century Vanguard Painting", which first appeared in (Duncan 1973) and then was revised for (Broude and Garrard 1982).

[18]   Smith had solo exhibitions in New York City in 1940 and 1943. In 1940, his work was shown at the Saint Paul (Minnesota) Gallery and School of Art, and in 1941 at the Kalamazoo (Michigan) Institute of Arts, and the Walker Art Center (Minneapolis). Skidmore College, Saratoga Springs held an exhibition of his art in 1942 (Lyon et al. 2021).

[19]   The sole exception seems to be Bradt's article where one of the two photographs illustrating the article showed Smith welding a piece of abstract sculpture. Bradt, "Few Welders".

[20]   "Welder-Sculptor", p. 76. This article appeared at the time of Smith's 1943 solo exhibition at the Willard Gallery. Smith's Schenectady address was 1113 McClellan Street (Falk 1991). According to Dehner, "We lived in an attic [in Schenectady] during that wartime housing shortage". Storm King Art Center curatorial files.

[21]   That fact is confirmed by Dehner's 1978 statement regarding a similar photograph of Smith with *Sewing Machine*: "It shows a sculpture David's which was largely carved at the marble works of Freeman LeBraque [sic] in Saratoga Springs where the usual produce was tombstones. David did it mainly with Freeman's power tools. He was always fascinated by industrial methods and had never hand carved marble, even though this picture depicts hammer and chisel". (AAA 3472-93). According to Smith, "when I was thru work at the factory at 8:00 A.M.–I would drive 2 or 3 days a week, 40 miles to Saratoga to the monuments works of Mallory and LaBrake where I carved marble for 6 h". Autobiographical notes, (Gray 1968).

[22]   The work is *Head as a Still Life II* (1942), *David Smith Sculpture: A Catalogue Raisonné*, #173.

[23]   "Welder-Sculptor", p. 76.

[24]   AAA 986-743, David Smith to Marian Willard, no date. "The N. Y. Office of American Locomotive got the ideas that [Smith] would be good publicity for them so they sent a Times Wide World Guy up here to take pictures one day…". AAA Dorothy Dehner Papers, Dorothy Dehner to Lucille Corcos and Edgar Levy, 1943. See also AAA 986-frame number illegible, Paul Rugile of Newsom to Marian Willard, 2 April 1943. For a typescript of the press release, dated 6 April 1943, see Estate of David Smith, Box 20, File Reviews.

[25]   One exception is a small photograph of Smith at work that accompanied (A Modern Metallurgist 1946).

[26]   For a complete list of Smith's exhibitions during this period, see *David Smith Sculpture: A Catalogue Raisonné*, I, 237–38 and 244–47.

[27]   One exception is the portrait of Smith posing with his models for the National Amateur Competition prize medal, which appeared in *Art News*, the sponsoring institution (Amateur Standing 1949).

[28]   AAA, AAUW Papers, Exhibitions, David Smith, 1946–1947.

[29]   (Greenberg 1952). As early as 1943, Greenberg had declared that Smith "has a chance of becoming one of the greatest of all American artists". "American Sculpture of Our Time: Group Show", *Nation* 156, no. 4 (23 January 1943): 140. Greenberg no longer felt the need to equivocate about Smith's superiority in 1956 when he referred to the artist as "the best sculptor of his generation" (*Art in America* 44, no. 4 (Winter 1956–1957): 30).

[30]   The most influential of these studies is (Guilbaut 1983). Kingsley's *Turning Point* is less thought-provoking in its approach but nonetheless reinforces this moment's importance for the New York School (Kingsley 1992).

[31]   The notion of a victory for these artists, with its obvious parallel to the recent military successes of the United States, was codified in the title of (Sandler 1970).

[32]   The perception that popular magazine's coverage of the Abstract Expressionists was uniformly negative in tone has been corrected by recent scholarship, especially by (Collins 1991). Collins revised that essay as (Collins 2020). Mary Corlett maintains the view that popular magazines were negative in their treatment of Abstract Expressionists, especially Pollock, although she differentiates between *Time* and *Life* and the more sympathetic coverage found in *Vogue*, *Harper's Bazaar*, and *New Yorker*, which catered to "a more specifically upper middle-class audience" (Corlett 1987). Bergstein notes the importance of mainstream American magazines for disseminating "The mystique of the artist's atelier", although she does not distinguish between the reporting or the intended audience of *Life*, *Vogue*, and *Harper's Bazaar* (Bergstein 1995).

[33]   Bergstein, "Artist", p. 54.

[34]   Narrative presentations of avant-garde artists at work were by no means absent in popular magazines. For example, Andrew Perchuk emphasizes the linearity with which Pollock is shown executing a painting in a series of Namuth photographs published as (Baffling U.S. Art: What is it About 1959). However, forging a connection to tradition was not the thrust of such a presentation, as Perchuk indicates (Perchuk 1995).

[35]   (de Kooning 1951) Elaine De Kooning was credited as the author of the article, but her text relied heavily on a typescript Smith prepared specifically for the occasion (Smith 2018).

[36]   For a discussion of the rebel in post-war American society, see (Landau 1989).

[37]   Brenson discusses this month in some detail, *David Smith*, pp. 517–25.

[38]   For example, the photograph on the upper left of page 24 of the Kramer article had, in 1938, appeared in *Magazine of Art*, credited to Leo Lances (Figure 3). Although not the topic of this study, Smith provided an important photographic record of his sculptures; see (Hamill 2015).

[39]   During the 1960s and the early 1970s, the audience for such images had expanded sufficiently to justify the publication of hardcover photoessays. Among the contributions of these two photographers to the genre were (Mulas 1967, 1971; Budnik 1970).

[40]   Bergstein discusses the growing importance of the relationship between artist and photographer in the modernist period. "The Artist in His Studio", p. 46.

[41]   On this episode of Smith's career, see (Smith and Carandente 1964).

[42]   On the presentation of Smith's sculpture in Spoleto, see (Sullivan 2013).

[43]   Smith and Carandente, *Voltron*.

[44]   "Goliaths", p. 129.

[45]   Marcus, *David Smith: The Sculptor and His Work*, p. 127.

[46]   For a fuller discussion of Smith's relationship with workers, see (Wisotzki 2005).

47  "Goliaths", p. 131.

48  Jones, "Clothes", p. 32.

49  De Kooning, "Makes a Sculpture", p. 38.

50  *Art in America*, 54 (January–February 1966): 48.

51  (Farewell to the Vulcan of American Art 1965) Smith's high level of productivity at Voltri had previously inspired Giovanni Carandente to exclaim "And Vulcan went to Voltri". *Voltron*, p. 5.

52  Most other American sculptors associated with direct metalworking techniques adopted them only in the 1940s. For example, Theodore Roszak took up welding as "a new sculptural approach" in (Arnason 1945).

53  An early and important discussion of this phenomenon is provided by (Rose 1980a).

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
