# Peer review of "More than a Man, Less than a Painter: David Smith in the Popular Press, 1938–1966"

_arts, 1938_

Round 1
Reviewer 1 Report
General comments
Overall a good review of several photographic representations of David Smith in media throughout his career, with well chosen examples. Generally well written and well referenced.
In general, the paper is largely descriptive and the overall/central argument/ original contribution to research is not entirely clear. The script could do with additional context/discussion in each section in order to orient the reader through the main ideas and contribution to scholarship. More in-depth critique is needed to fully examine the examples provided.
The paper would benefit from a much stronger conclusion that sums up the argument and outlines its main points.
Futher comments
p0: '...recent scholarship'. Consider that Kraus's work is not particularly 'recent'
p4: Important to refer to the fact that Smith made the first welded steel sculpture in America, although PIcasso and Gonzales had created welded steel work in 1927/28 in Europe.
p4: '...the low trajectory of his career...' Consider rephrasing here. Also perhaps noteworthy that this was in the middle of the depression the years spent as technical director at the FAP for the WPA, and the fact that Smith and Dehner spent 1935-36 travelling in Europe.
p4: '... served as little more than announcements for his exhibitions..' : Citation needed.
p7: 'artists inventive finishes...': codsider revising this paragraph or making the point more clearly. It is not clear at all what the point is here, or how 'unusual facts and ideas' in the publication is in opposition/ contrast with 'shaping the female form'?
p9. Consider rephrasing: It is highly likely that most photographs of artists at work at this period were staged. Perhaps also important to point the reader to the most notable example of this in Hans Namuth's contrived film footage of Pollock at work, filming the artist from below a sheet of glass - famoulsy hated by Pollock - there are other examples of artists at work being staged in the literature.
p9: '...staging his process fdor the photography..' Consider that the photographer is likelty to have directed the 'staging' of the scene, rather than the artists. Again, there are examples in the literature. If it is important that this point is made, then provide a citation/example of where an artist staged a scence for a photo shoot.
p12. 'with the aid of an air compressor'. While this is explained further in the end note, consider adding some context here in the paragraph. The use of an air compressor is not necessarily evidence of an 'industrial' method. Consider adding some of the context in the footnote to the main text. eg. Smith used industrial carving techniques, as this was more efficient then learning stone carving from scratch. This is a more convincing example to connect direct carving to the modern industrial methods Smith took from his early factory experiences and having shared space with factory workers at Terminal Iron Works in Brooklyn.
p15: 'photos were intended to establish legitimacy ...' This needs careful revision or rephrasing. It is not entirely clear that this was the intent of the photographer/publication when highlighting Smith's work and process. Consider that the publications may have simply been interested in a novel approach to art making that might be of interest to their readers.
p15: "...as a skilled armor plate welder.": Consider removing the direct quote here, as it doesn't particularly add anything to the point. Paraphrasing the quote might be better.
p16: '...years passed without publiction of a photograph...' This is stated without context. Speculate as to why this was.
p19: Pollock might be introduced here, as the most famous photographed figure of all representations of AE artists at this time, rather than introducing this at the end of the paper.
Also here consider adding a reference to the fact that Smith obsessively photographed his own work in various locations, and even himself in his own studio at Bolton Landing. Hamill's work discusses this at length. It would form an important contrast with the illustrations in the paper of how Smith and his work were illustrated in publications.
p22: '...with credit line of professional photographers...': Citation/example needed.
p22: 'Bolton Landing'. Add 'New York'. Anyone unfamiliar with Smith will not know where BL is situated.
p22: Mention of the Festival of the Two Worlds at Spoleto: Citation needed.
p22: 'Industrial detritus...': Consider rephrasing. The materials Smith used at the Voltri factories were largely old machine parts, tools, spare parts and steel remnants, rather than detritus.
p23: '...Caradente placed sculptures...': Citation needed.
p23: 'capturing more thoughtful images of the artist': There are several earlier examples of Smith pondering and thoughtulf from Bolton Landing and elsewhere.
p23: '...as macho artist...': Citation/examples needed of other images that demonstrate /illustrate this.
Note also: while Smith is often described as an imposing figure etc. he was not exceptionally tall at 6'2".
p30: Argument that Smith is a barrier between the viewer and the sculptures. Not entirely convincing as is. Consider added a little more discussion here.
p30: '...replaced by the artist as a force of nature with god-like creative abilities...' This is perhaps a little too much hyperbole. Consider rephrasing. Or add additional context to the discussion of the image.
p31: '...Arshile Gorky, Pollock and Mark Rothko'. Inconsistency: Either use artist surname OR first name/surname.
Author Response
Page 0 eliminated recent
Page 4 added information about Picasso and Gonzales in a new footnote.
Page 4 still believe the phrase low trajectory in connection with other modernist artists is appropriate but added information about Smith's "day job" in a foot note.
Page 4. Added citation documenting short announcements without illustrations from 1938-1940.
Page 7. reworked the analysis of the Popular Science page.
Page 9 (both comments). reworked text and note to acknowledge Pollock's images as staged.
Page 12. corrected to industrial carving technique. The point for me is really the difference between the execution of the work and the published images of him with them. Therefore, I did not expand the discussion of the making of the stone works of the early 1940s.
Page 15. I reworded the phrase in question.
Page 15. I think the quotation is useful because of its specificity.
Page 16, The next paragraph goes on to explain at length my argument for why those images disappear in the late 1940s.
Page. 19. As to when I choose to introduce Pollock -- that's how I chose to handle the material. I think it works.
I did add a reference to Smith's photographs of his own works (which of course are also mentioned in the introductory paragraph).
Page 22. I provided a citation of an example.
Page 22. Added, New York
Page 22. Added citation for Spoleto festival.
Page 22. rephrased as requested.
Page 23. Added citation for Spoleto installation.
Page 23. I disagree that there are other (than those I discuss) images published during Smith's lifetime. The average american male was 5'7" (146 lbs) in 1950 and 5'8" (160 lbs) in 1960 so I feel perfectly justified in saying Smith was unusual at 6'2" and 220 lbs.
Page 30. I stand by my analysis of the photograph and the aura that has sprung up around it as a memorial tribute to Smith.
Page 31. I've given Pollock's full name previously in the text so will not provide it at the secondary instance.
Reviewer 2 Report
The submission should address this source: https://yorkspace.library.yorku.ca/xmlui/handle/10315/29883
A number of arguments made in this submission were made in this earlier dissertation. Furthermore, there's a section on David Smith and the popular press and enough overlap in topic. Author needs to distinguish their topic more and acknowledge the prior research on David Smith in the popular press.
Author Response
I have acknowledged Bissonnette's dissertation along with documenting my development of this project. In addition, I cite my 2005 publication which addresses images of Smith and their political implications which also significantly predates her dissertation.
Reviewer 3 Report
This is an interesting topic that was traditionally the subject matter of eighteenth-century portraiture and now is the subject of professionalism and self-presentation in the arts and media. The Abstract Expressionists were early examples of artists well-lauded and in conflict with self-presentation, publicity, and the press. The author avoided obvious examples such as the Irascibles photo shoot and the Hans Namuth Pollock painting photographs and film. These should be mentioned in the footnotes. Also particularly nice would be their set-up of the presentation of sculpture at Bolton Landing as opposed to David's Smith self-presentation as the artist himself. Krauss's recollection of visiting the space and talking to Clement Greenberg would also be a cohesive mention.
Author Response
Since Smith was not part of the "Irascibles" photo shoot, it is not relevant to this material. Another version of this material which addresses Smith's self-presentation without relying strictly on photographs published during Smith's lifetime, addresses working man vs. working artist and therefore takes into account B. Newman's "plan" for the Irascibles photo shoot, but again, it's not relevant here. And, since I'm not investigating the presentation of the sculpture, but instead the presentation of Smith working (as Reviewer 3 states) -- a more generalized discussion of the sculpture at Bolton Landing is irrelevant.
Round 2
Reviewer 2 Report
- Being able to see reproductions of these historical publications on David Smith, alongside the author's analysis, provides a much-needed addition to the literature on Smith. They allow readers to see Smith's work in its original context, and to appreciate the evolution of his ideas.
- Page 12, second paragraph, line 3 - spelling error in title: “Sculptor, Alco Employe.
- The switch between double hyphens (--) and em dashes (—) is distracting and should be made consistent.